# *Clostridioides Difficile* Infection before and during Coronavirus Disease 2019 Pandemic—Similarities and Differences

**DOI:** 10.3390/microorganisms10112284

**Published:** 2022-11-17

**Authors:** Nadica Kovačević, Vedrana Petrić, Maria Pete, Milica Popović, Aleksandra Plećaš-Đurić, Slađana Pejaković, Slavica Tomić, Dimitrije Damjanov, Dijana Kosijer, Milica Lekin

**Affiliations:** 1Faculty of Medicine, University of Novi Sad, 21137 Novi Sad, Serbia; 2Clinic for Infectious Disease, University Clinical Center of Vojvodina, 21137 Novi Sad, Serbia; 3Clinic for Nephrology and Clinical Immunology, University Clinical Center of Vojvodina, 21137 Novi Sad, Serbia; 4Clinic for Anesthesiology, Intensive Care and Pain Therapy, University Clinical Center of Vojvodina, 21137 Novi Sad, Serbia; 5Clinic for Endocrinology, Diabetes and Metabolic Disorders, University Clinical Center of Vojvodina, 21137 Novi Sad, Serbia; 6Clinic for Gastroenterology and Hepatology, Clinical Center of Vojvodina, 21137 Novi Sad, Serbia

**Keywords:** *Clostridioides difficile* infection, COVID-19, risk factors, outcome

## Abstract

The aim of this study was to investigate the differences of *Clostridioides difficile* infection (CDI) during the COVID-19 pandemic compared to the pre-COVID-19 era. CDI patients treated at the Clinic for Infectious Diseases, Clinical Center of Vojvodina, Serbia during 2017–2019 (*n* = 304) were compared with COVID-19/CDI patients treated in period September 2021–September 2022 (*n* = 387). Groups were compared by age, gender, comorbidities, previous medications, laboratory findings, and outcome within 30 days. In the CDI/COVID-19 group, we found: greater percentage of males 59.8% vs. 42.6% (*p* ≤ 0.001), older age 72.8 ± 9.4 vs. 65.6 ± 11.7 (*p* ≤ 0.001), higher Charlson comorbidity score (CCS) (3.06 ± 1.54 vs. 2.33 ± 1.34 (*p* ≤ 0.001), greater percentage of chronic renal failure (33.9% vs. 23.4% (*p* = 0.003), malignances (24.3% vs. 13.5% (*p* ≤ 0.001), chronic obstructive pulmonary disease (22.7% vs. 15.5% (*p* = 0.017), higher usage of macrolide (38.5% vs. 8.6% (*p* ≤ 0.001), greater percentage of patients with hypoalbuminemia ≤25 g/L (19.6% vs. 12.2% (*p* ≤ 0.001), lower percentage of patients with elevated creatinine (≥200 mmol/L) (31.5% vs. 43.8%) (*p* = 0.002), and greater percentage of lethal outcome 29.5% vs. 6.6% (*p* ≤ 0.001). In the prediction of lethal outcome multivariate regression analysis extracted as an independent predictor, only higher CRP values in the non-COVID-19 group and in the COVID-19 group: older age (*p* ≤ 0.001), CCS (*p* = 0.019) and CRP (*p* = 0.015). COVID-19 changes the disease course of CDI and should be taken into consideration when managing those patients.

## 1. Introduction

The ongoing pandemic of Coronavirus disease 2019 (COVID-19) has reduced the importance of many other infectious diseases in the last three years. The multisystemic nature of the disease has indicated the need for uniting doctors of different specialties into a united team. The explanation of the wide range of diverse clinical manifestations of the disease is based on the specific properties of the SARS-CoV-2 virus, which infects host cells by binding its spike glycoprotein (S) to angiotensin-converting enzyme 2 (ACE2), which is a functional receptor in various tissues of the human body. It is found in the oral and nasal mucosa, nasopharynx, lungs, kidneys, brain, heart, spleen, liver, stomach, small intestine, colon, lymph nodes, thymus, and bone marrow. The SARS-CoV-2 virus mostly binds to ACE2 receptors in alveolar epithelial cells of the lungs, as well as endothelial and muscle cells of arteries and veins in all organs [1,2]. Apart from respiratory symptoms, which are the most common, some patients (2–50%) also have diarrhea as part of the clinical presentation [3]. Initially, COVID-19 pneumonia is viral and interstitial by its nature, but in the further course, it can be complicated by bacterial superinfection, which increases the need for the introduction of antibiotic therapy, thereby putting the patient at risk of developing unwanted complications such as *Clostridioides difficile* infection (CDI) [4]. On the one hand, infection with the SARS-CoV-2 virus leads to damage to the gastrointestinal mucosal barrier of the host, and on the other hand, a disruption of the intestinal microbiota occurs. These are the main predisposing factors for the development of CDI in COVID-19 patients. The use of antimicrobials that are applied during the treatment of patients with COVID-19 additionally contributes to the harmful effect on the gut microbiota [5,6,7]. The immune system plays an important role in the clinical manifestation of CDI. Immunological studies conducted in the prepandemic period proved that patients with sufficient production of antitoxic antibodies have a lower probability of developing a severe form of the disease and CDI relapse. Research conducted during the pandemic has further established that a disruption in the immune response in COVID-19 patients may also influence the occurrence of CDI [8]. If the immune response of the infected host is too extensive, it leads to a hyperinflammatory syndrome called a “cytokine storm”, which is characterized by extreme concentrations of proinflammatory cytokines and leads to multiorgan damage. Research has shown that the cytokines detected in patients with severe forms of COVID-19 (IL-1β, IL-6, IL-8, IL-17A, IL-16) are very similar to the immune production of cytokines in patients with CDI. An increase in the production of Th17, IL-6, and IL-23 is associated with more severe forms of CDI and death in patients with CDI [9,10,11]. Epidemiological studies in recent years have recorded an increase in the incidence, severity of the clinical presentation, and mortality rate of CDI, while an unacceptably high rate of CDI relapse (10.1–50.8%) is still registered despite the currently available therapy [5,12,13]. These epidemiologic changes are mainly attributed to the biological characteristics and pathogenicity of new virulent CDI strains and the aging population in many countries [14]. Although great efforts have been made in the past decade to address many aspects of CDI, there are still unresolved problems such as underdiagnosis and inadequate use of antibiotics, which have even worsened during the COVID-19 pandemic.

At the beginning of the pandemic, the results of the studies indicated a reduced incidence of CDI in COVID-19 patients, primarily due to prevention measures aimed at preventing the spread of the SARS-CoV-2 virus, but also due to insufficient diagnostics because CD-induced diarrhea was interpreted as part of the clinical presentation of COVID-19. Delayed diagnosis of CDI puts the patient at risk of developing more severe forms of the disease, especially in vulnerable populations such as the elderly, immunocompromised individuals, and individuals with multiple comorbidities in whom there is a need for prolonged and repeated hospitalization [2,6,15,16]. However, some recent studies have indicated that the incidence rate of CDI increased significantly from 2.6% in the prepandemic period to 10.9% during the COVID-19 pandemic [17,18]. Moreover, the study of Lewandowski and coworkers showed that COVID-19-related involvement of the gastrointestinal tract is an independent risk factor for CDI, most probably because of the alteration of the gut microbiome caused by SARS-CoV-2 [17].

Changes in the epidemiological, immunological, and clinical picture in recent years have made the CDI of today different from the CDI of the past [6,18,19]. Continuous exposure to antibiotics has caused CD to develop into a multidrug-resistant (MDR) pathogen with increasing virulence, and around 60% of clinical CD strains are already reported as MDR in European hospitals [5,20]. For the abovementioned reasons, CDI is today considered one of the most important nosocomial infections worldwide and one of the most important threats to global health [6,18,19]. Increasingly, *C. difficile* infections demonstrate an emerging pattern of resistance to available treatment and recurrence after an initial episode [21]. The fact that the growing rate of morbidity, relapse, and mortality of CDI also leaves significant economic implications for the healthcare system should also be taken into consideration.

The aim of this research was to compare the epidemiological and clinical characteristics and outcomes of hospital-acquired CDI (HA-CDI) in patients in the prepandemic period and during the COVID-19 pandemic. 

## 2. Methods

We conducted a retrospective single-center analysis that included a total of 791 patients with confirmed CDI, aged ≥18 years. The first group consisted of 387 COVID-19 patients with CDI treated at the COVID-19 Hospital at the University Clinical Center of Vojvodina, Serbia, from 2 September 2021 (when this hospital was opened) to 2 September 2022. The second group included 304 patients with confirmed CDI, hospitalized at the Clinic for Infectious Diseases at the University Clinical Center of Vojvodina, Serbia, in the period before the pandemic, from January 2017 to December 2019.

The COVID-19 Hospital is a tertiary healthcare institution within the University Clinical Center of Vojvodina, where only patients with COVID-19 are hospitalized since 2 September 2021. 

We compared the epidemiological and clinical characteristics and outcomes of hospital-acquired CDI (HA-CDI) in patients treated in the prepandemic period (January 2017–December 2019) and the COVID-19 period (September 2021–September 2022).

Demographic, epidemiological, and clinical characteristics of COVID-19 and CDI, comorbidity status, antimicrobial therapy before the diagnosis of CDI and during the treatment of CDI (antibiotics, proton pump inhibitors–PPI, corticosteroids), laboratory parameters (values of leukocytes, C-reactive protein, albumin, and creatinine) and disease outcome in the 30-day follow-up period after diagnosis. The data was collected from the electronic medical records of hospitalized patients.

The diagnosis of COVID-19 was confirmed by real-time polymerase chain reaction (PCR) from nasopharyngeal and oropharyngeal swabs. The case confirmation was obtained using the Rotary Nucleic Acid Extraction System (GeneRotex 96L) (Xi’an Tianlong Science-Medicina 2022, 58, 1262 3 of 11 and Technology Co., Ltd., Xi’an, China) and the Gentier Real-Time Quantitative PCR (Gentier 96E) (Xi’an Tianlong Science and Technology Co. Ltd., Xi’an, China). The clinical form of COVID-19 infection was defined in accordance with World Health Organization (WHO) criteria as follows: mild pneumonia—clinical signs of pneumonia (fever, cough, dyspnea, and fast breathing) but no signs of severe pneumonia, including an SpO_2_ of ≥90% on room air; and severe pneumonia—clinical signs of pneumonia (fever, cough, dyspnea, and fast breathing) plus one of the following: a respiratory rate of >30 breaths/min, severe respiratory distress, or an SpO_2_ of <90% on room air, and increased inflammatory markers [22]. 

The diagnosis of CDI was based on the presence of diarrhea (≥3 watery stools within 24 h) associated with detection of the *C. difficile* toxins. Because no single test is suitable to be used as a stand-alone test, use of a 2-step testing algorithm is recommended by European guidelines. Algorithms currently recommended by the ESCMID comprise a screening test with high sensitivity followed by a more specific test to detect free toxins. In this approach, stool is first tested using a highly sensitive GDH test, and the second test is the more specific toxin EIA. If both are positive, the diagnosis of CDI can be made reliably [23]. In our study, the etiology was confirmed by the enzyme-linked fluorescent assay (ELISA) and the RIDASCREEN C. difficile Toxin A and B (C0801), R-Biopharm AG, Germany. The testing was performed via glutamate dehydrogenase (GDH) and ELISA for the toxin and was considered diagnostic if both tests were positive. 

CDI severity was defined in accordance with the European Guidelines for the treatment of CDI as follows: mild CDI—absence of the following criteria: fever (>38.5 °C), hemodynamic instability, leukocytosis (leukocytes > 15,000 cells/µL), serum creatinine increase of >1.5 times the values before infection, increase in serum lactates, histological evidence of pseudomembranous colitis, and radiological evidence of ileus or ascites; severe CDI—the presence of at least one of the following criteria: fever (>38.5 °C), hemodynamic instability, leukocytosis (leukocytes > 15,000 cells/µL), serum creatinine increase of >1.5 times the values before infection, increase in serum lactates, histological evidence of pseudomembranous colitis, and radiological evidence of ileus or ascites; and complicated CDI—an episode of CDI complicated by toxic megacolon, intensive care unit hospitalization, sepsis, surgery, or death caused by CDI [24]. Hospital-onset CDI (HO-CDI) was considered if the symptom onset was >72 h from hospital admission. 

The study was approved by the Ethics Committee of the Hospital (Nº 00-100/13.05.2022) IRB00013195 Ethics Committee of Serbia IRB#1. The study was conducted in accordance with the Declaration of Helsinki.

Statistical analysis—Data were analyzed using SPSS v. 23.0 software. Categorical variables were shown as number (percentage), while mean values of continuous variables were given as arithmetic means and standard deviations, after the check of the distribution. Differences between groups were compared by chi-squared test for the categorical variables and by *t*-test for the continuous ones. For the purpose of defining independent predictors of lethal outcome we performed multivariate regression analysis using binary logistic regression model. We constructed regression model with the inclusion of the lethal outcome as the dependent variable. For the independent variables, we picked those variables that showed *p* value less than 0.05 using univariate analysis. In the case of multicollinearity between parameters, we included in the model the parameter that showed greater predictive value using univariate analysis. Significance (*p*) was set at the value of 0.05.

## 3. Results

The results of the comparison of CDI in the pre-COVID-19 period with coinfection of COVID-19 and CDI are shown in Table 1. 

Patients with CDI/COVID-19 coinfection are older, 72.8 ± 9.4 vs. 65.6 ± 11.7 (*p* ≤ 0.001). Charlson comorbidity index is a list of 19 clinical conditions that are individually scored 1–6. Charlson score represents the summarized value of existing comorbid conditions in the observed patient [25]. CDI/COVID-19-coinfected patients had higher Charlson comorbidity score, 3.06 ± 1.54 vs. 2.33 ± 1.34 (*p* ≤ 0.001). In this group, a higher frequency of patients with chronic renal failure 33.9% vs. 23.4% (*p* = 0.003), malignancy, 24.3% vs. 13.5% (*p* ≤ 0.001), and chronic obstructive pulmonary disease (COPD), 22.7% vs. 15.5% (*p* = 0.017), was recorded. Of the antibiotics used before the onset of CDI, a higher use of macrolides was recorded 38.5% vs. 8.6% (*p* ≤ 0.001). Laboratory findings show a higher percentage of patients with severe hypoalbuminemia (less then 25 g/L), 19.6% vs. 12.2% (*p* ≤ 0.001) and a lower percentage of those with creatinine over 200, 31.5% vs. 43.8% (*p* = 0.002). By analysis of concomitant therapy, all drugs of importance (antibiotics, proton pump inhibitors, chemotherapy, and corticosteroid therapy) were more frequently used in the CDI/COVID-19 group. In this group of patients, a significantly higher mortality rate was recorded 29.5% vs. 6.6% (*p* ≤ 0.001). Because of the significant differences between the observed groups, the risk factors for the fatal outcome were examined individually in each of the patient groups.

**Prepandemic CDI group**—The results of the univariate analysis comparing deceased and surviving patients in the pre-COVID-19 group are shown in Table 2. Deceased patients had higher frequency of diabetes mellitus, 60% vs. 50% (*p* ≤ 0.001), and higher Charlson comorbidity score, 2.90 ± 1.45 vs. 2.29 ± 1.33 (*p* = 0.049). They had greater number of antibiotics used before CDI onset, 1.85 ± 0.74 vs. 1.06 ± 0.48 (*p* ≤ 0.001), greater percentage of patients treated with cephalosporins of the III generation, 80% vs. 43.2% (*p* ≤ 0.001), and fluoroquinolones, 65% vs. 37.2% (*p* = 0.048), than recovered patients. Moreover, the deceased group of patients were in higher percentage treated with PPI (*p* = 0.023) and corticosteroids (*p* = 0.015). All deceased patients had a WBC number greater than 15 × 10^9^/L. Thirty percent of deceased patients had hypoalbuminemia less than 25 mg/L, while in the recovered group, that percentage was only 10% (*p* = 0.012). CRP was significantly higher in the deceased group of patients, 288.65 ± 48.77 vs. 116.67 ± 69.02 (*p* ≤ 0.001).

Multivariate regression analysis singled out CRP as the only variable with an independent contribution to the prediction of lethal outcomes (Table 3). The distribution of CRP values among deceased and surviving patients is shown in Chart 1.

**COVID-19/CDI-coinfected group of patients**—The results of the univariate analysis comparing deceased and surviving patients in the group of coinfected with COVID-19 and CDI are shown in Table 4. Deceased patients were significantly older, 80.2 ± 6.8 vs. 70.0 ± 6.8 (*p* ≤ 0.001), with a higher Charlson comorbidity score, 3.95 ± 1.45 vs. 2.70 ± 1.42 (*p* ≤ 0.001), and a higher percentage of patients with cardiovascular disease, 48.2% vs. 33.7% (*p* = 0.007), diabetes mellitus, 66.7% vs. 49.5% (*p* = 0.002), gastrointestinal tract diseases, 32.5% vs. 19.4% (*p* = 0.006), chronic renal failure, 53.5% vs. 25.6% (*p* ≤ 0.001), neurological disease, 34.2% vs. 11.7% (*p* ≤ 0.001), and COPD, 35.1% vs. 17.6% (*p* ≤ 0.001). The number of antibiotics used, 1.94 ± 0.93 vs. 1.26 ± 0.76 (*p* ≤ 0.001), as well as the percentage of patients with previous usage of cephalosporins of the III generation, 71.1% vs. 36.6% (*p* ≤ 0.001), and fluoroquinolones, 60.5 vs. 38.8 (*p* ≤ 0.001), were higher in the group of deceased patients, while amoxicillin clavulanic acid was more used in patients who survived (*p* = 0.024). All relevant laboratory parameters (WBC greater than 15 × 10^9^/L, hypoalbuminemia less than 25 g/L and creatinine greater than 200 mmol/L) were more frequent among the deceased patients (*p* ≤ 0.001) in the univariate analysis.

Multivariate regression analysis identified age, Charlson comorbidity score, and CRP as independent predictors of the lethal outcome (Table 5). The distribution of patient age, Charlson comorbidity score, and CRP values among deceased and surviving patients is shown in Charts 2–4.

## 4. Discussion 

*Clostridioides difficile* infection is today considered one of the most significant hospital infections worldwide, and the greatest risk for the development of this infection is in elderly hospitalized patients with multiple comorbidities. During the COVID-19 pandemic, the caution regarding the possible increase in the incidence and mortality of HA-CDI has increased, especially in the high-risk patient population. The reasons for this concern are based on the increased number of people hospitalized during the pandemic and the high consumption of antibiotics recorded in many hospitals in the last three years. Prolonged and repeated hospitalizations and the use of broad-spectrum antibiotics are well-known triggers of CDI in hospitals [14,26].

In our study, we compared the characteristics of patients with CDI during the COVID-19 pandemic and the characteristics of patients with CDI before the pandemic. We were interested in what changed in terms of CDI during the pandemic and what remained similar.

Two groups were compared by age, gender, comorbidities, previous medications usage, laboratory findings, and outcome within 30 days period after diagnosis of CDI. In each group separately, risk factors for the lethal outcome were evaluated by multivariate binary logistic regression analysis.

An analysis of the demographic characteristics of our patients showed results similar to the majority of previous studies, in which the largest number of CDI patients were of older age [6,17,19]. In our study, patients with CDI/COVID-19 coinfection were older compared to CDI patients in the prepandemic period, 72.8 ± 9.4 vs. 65.6 ± 11.7 (*p* ≤ 0.001). The results of our study confirmed the conclusion of previous research that the severity and frequency of CDI have increased rapidly in recent years, primarily in the population of patients older than 65. This trend can partially be explained by the decreased function of the immune system, which is why most people of this age are considered immunocompromised, to have a decrease in the gastric acidity, a decrease in the protective function of bifidobacteria in the intestinal tract and have more frequent exposure to *C. difficile* due to repeated hospitalizations and multiple comorbid conditions. The clinical and epidemiological significance of the development of CDI in the elderly population is today reflected in the increasingly frequent occurrence of recurrent forms of the disease, poorer response to therapy, and an increase in the number of complications and mortality rate [17,18,19]. In some studies, the age structure of patients did not differ in the pandemic and prepandemic period [17].

Concerning the gender, in the group of patients with CDI/COVID-19 coinfection, a higher number of male patients was registered compared to the prepandemic period (59.8% vs. 42.6%) (*p* ≤ 0.001). Similar results published by Lewandowski et al. show that the representation of men with CDI in the pandemic period was 45.8% and in the prepandemic period 28.6% (*p* = 0.049) [17]. Such results could be interpreted by the fact that, during the COVID-19 pandemic, this viral infection was more common in men. Nevertheless, the research conducted by Vasquez-Cueste et al. did not register a significant difference in gender between the studied groups of patients [19].

Charlson comorbidity score (CCS) represents the sum of various comorbid conditions that significantly helps in identifying patients who are at increased risk for developing more severe forms of CDI [25]. Hardt showed this in his research conducted in the prepandemic period, in which patients with a milder form of CDI had a statistically significantly lower CCS compared to patients with a more severe form of CDI (3.4 ± 2.2 vs. 5 ± 2.6 *p* < 0.001) [27]. In our study, the average value of CCS was significantly higher in CDI patients with COVID-19 coinfection compared to CDI patients in the prepandemic period (3.06 ± 1.54 vs. 2.33 ± 1.34) (*p* ≤ 0.001). Similar to our research, Vasquez-Cueste et al. found a significant difference in the CCS value in the COVID-19 and the non-COVID-19 group (*p* = 0.034) [19]. The result obtained is explained by the fact that hospitalized patients with COVID-19 viral infection had multiple comorbidities, which further predisposed them to the development of CDI.

In our research, in the matter of particular comorbidities, a significantly higher percentage of patients with chronic renal failure, malignancies, and chronic obstructive pulmonary disease was recorded in patients with CDI/COVID-19 coinfection compared to CDI patients in the prepandemic period (33.9% vs. 23.4% *p* = 0.005; 24.3% vs. 13.5% *p* ≤ 0.001; and 22.7% vs. 15.5%, *p* = 0.017, respectively). In his study, Lungulescu registered malignancy in 46.8% of patients with a more severe clinical picture and in 26.1% of patients with a milder form of CDI, which indicates the fact that patients with malignancy develop more severe forms of the disease statistically significantly more often (*p* < 0.001) [28]. Research conducted in the prepandemic period also proved that patients with malignancies have a worse clinical response to the applied therapy [29]. Contrary to the previous ones, the research of Vasquez-Cueste et al. showed that patients with CDI and COVID-19 had a lower incidence of malignancy than the non-COVID-19 group. In the aforementioned study, the number of immunosuppressed patients was also lower in the non-COVID-19 group (*p* = 0.024) [19]. Regarding the patients with chronic kidney diseases, Lewandowski et al. developed a conclusion similar to our results that chronic kidney diseases and diseases of the nervous system were significantly more common in CDI patients during the pandemic than in CDI patients before the pandemic (31.3% vs. 15.6%., *p* = 0.038 and 39.6% vs. 11.7%, *p* ≤ 0.001 for nervous system disease) [17]. According to data published by Lis et al., risk factors that significantly contribute to the increase of CDI in patients with chronic renal failure are the advanced stage of chronic kidney disease, the length of antibiotic use, as well as lower albumin concentration [30].

Research indicates that 70–90% of CDI cases occur after the use of antibiotics, and the risk of developing CDI increases with multiple and prolonged use of antibiotics [6]. The use of antibiotics represents the most striking difference between the prepandemic and the pandemic period. According to published research, 87.5% of patients with CDI/COVID-19 coinfection were treated with antibiotic therapy during the pandemic, which is significantly more compared to the prepandemic period, in which 67.5% of patients received antibiotic therapy before the onset of CDI (*p* = 0.012) [17]. Our results confirm the conclusions of many studies about an extremely high percentage of antibiotic use before the onset of CDI both during the pandemic and in the prepandemic period. The situation was similar in many other countries [6,17,31,32]. In the pandemic era, macrolides, third-generation cephalosporins, and quinolones were the most commonly prescribed antibiotics for the treatment of COVID-19 in our patients as empiric therapy for potential bacterial superinfection of the respiratory tract. A similar situation was reported in other studies as well (31). In the prepandemic period, third-generation cephalosporins (45.1%) and quinolones (39.1%) were most often used in our patients before the onset of CDI. The only significant difference between the COVID-19 and non-COVID-19 groups of patients exists in the use of macrolides, which were significantly more used in the COVID-19 pandemic (38.5% vs. 8.6%, *p* ≤ 0.001). This is also confirmed by other studies regarding the antiviral and immunomodulatory activity of azithromycin [6,17,33]. An encouraging conclusion that the administration of azithromycin did not significantly affect the occurrence of CDI in patients with COVID-19 viral infection came from Lewandowski et al. during the pandemic [17]. Research by Vasquez-Cueste and Manea et al. also showed that the use of third-generation cephalosporins was significantly higher in the group of patients with COVID-19 than in the groups without COVID-19 [19,33]. In our research, PPI and corticosteroids were significantly more used in the COVID-19 pandemic, which is not surprising since these drugs are part of the protocol for the treatment of COVID-19 infection (Table 1).

The laboratory picture of CDI is often characterized by hypoalbuminemia. CD toxin A increases the vascular and mucosal permeability of the intestinal tract, which results in the intraluminal accumulation of liquid rich in serum albumins. In our patient population, hypoalbuminemia (less than 25 g/L) was statistically significantly more common in patients with CDI/COVID-19 coinfection (19.6% vs. 12.2%, *p* = 0.004). Low albumin values represent a marker of long-term associated chronic diseases, poor nutritional status, and poor immune function of the host, and therefore insufficient production of antitoxic IgA antibodies to *C. difficile*. Some studies even support the view that a ten-day or two-week antimicrobial therapy of the first episode of CDI in patients with more pronounced hypoalbuminemia is not sufficient to eliminate the infection, and that a prolonged therapeutic regimen of vancomycin should be used in these patients in the first episode of CDI [34]. After CD colonization, the immune response of the colonized person plays an important role in the further course and outcome of CDI. In the prepandemic period, the results of several studies showed that CRP as a marker of inflammatory response is a statistically significant predictor for the development of more severe forms of CDI [27,35]. Solomon et al. have proven that infection with highly virulent strains of CD that produce a larger amount of toxins is accompanied by a stronger inflammatory response and that there is a positive correlation between the concentration of CD toxins and the values of leukocytes and CRP in the patient’s blood. Namely, the patients with the highest concentration of CD toxin also had a severe form of CDI, with CRP values over 250 mg/L and leukocytes over 20 × 10^9^/L with reduced blood albumin values [35]. The importance of determining biomarkers in the blood of patients with CDI was also confirmed by Herbert et al., who showed that CRP values were statistically significantly higher in patients with CDI compared to patients with CD-toxin-negative diarrhea (126 mg/L vs. 58.5 mg/L, *p* = 0.001) and that albumin values were statistically significantly lower in patients with CDI compared to patients with CD-toxin-negative diarrhea (22 g/L vs. 25 g/L, *p* = 0.003) [36]. The results of the mentioned studies point to the fact that elevated values of inflammatory markers in patients with diarrhea of unclear etiology should certainly arouse suspicion of the possibility of CDI. However, we did not observe significantly higher CRP values before and during the COVID-19 pandemic (128.0 ± 80.14 vs. 133.4 ± 75.24) (*p* = 0.357).

**CDI OUTCOME**—In coinfection with CD and the SARS-CoV-2 virus, worse outcomes and increased mortality can be expected, given that research during the pandemic proved that CD stimulates the production of inflammatory cytokines, which are very similar to the production of cytokines in patients with a more severe form of COVID-19 [5,6,36]. Mortality rates averaged 5–17% in studies conducted before the pandemic [37,38]. In the pandemic era, most authors published significantly higher mortality rates in patients with CDI and SARS-CoV-2 coinfection (22.5%, 19%, 44%, 28.9%) [15,31,32,39]. Our data also indicate significantly higher mortality in patients with CDI/COVID-19 coinfection compared to mortality in CDI patients before the pandemic (29.5% vs. 6.6%, *p* ≤ 0.001). This conclusion was also published in the research by Vasquez-Cueste et al. [19]. Some studies even indicated a lower incidence of CDI in the pandemic compared to the prepandemic period, but, at the same time, a worrying increase in the mortality rate due to CDI was observed [40,41]. Therefore, clinicians and epidemiologists have shown considerable interest in potential risk factors for mortality in CDI in recent years. Meta-analyses that dealt with mortality in CDI patients showed that age >65 years, leukocytes >20 × 10^9^/L, creatinine >200 mmol/L, and serum albumins <25 g/L represent significant risk markers for mortality. The authors emphasized that these parameters are very useful for mortality risk assessment because they are inexpensive, objective, and can be easily registered in the early stages of CDI, which is very important for the further course of the disease and the application of appropriate therapy [42]. Studies by Solomon et al. have shown that there is a significant correlation between the concentration of CD toxins in the patient’s blood and the values of C-reactive protein, leukocytes, and albumin. The authors of the study concluded that patients infected with CD strains that produce a high toxin titer have significantly elevated levels of inflammatory markers in the serum, including leukocytosis > 20 × 10^9^ /L and CRP > 230 mg /L, which may be useful risk markers for mortality in CDI [35].

In the prediction of a lethal outcome multivariate regression analysis in our research extracted only higher CRP values as an independent predictor in the non-COVID-19 group, while independent predictors in the COVID-19 group were older age (*p* ≤ 0.001), CCS (*p* = 0.019), and CRP (*p* = 0.015). Age is a well-known risk factor for severe CDI forms and poor outcomes due to a weaker immune response to *C. difficile* toxins and the presence of more chronic diseases. Mortality increases with age, and according to numerous studies, patients aged over 65 are at particularly high risk for mortality [38,40,42]. In Italy, a mortality rate of 10.6% was reported for patients aged 60–69 and 31.7% for patients aged 80–89 during the COVID-19 pandemic [43].

In our study, the average age of deceased patients with CDI/COVID-19 was 80.2 ± 6.8 years. Similar to our results, in the research of Bednarska et al., the average age of people who died with CDI was 83 years, and the authors showed that an age over 77 represents an indicator of increased risk of death in CDI patients [41]. Similar results were presented by Czepiel et al., who registered an average age of 80 years in deceased patients. These patients were, on average, 8 years older than the surviving CDI patients (*p* < 0.001). In addition to age, the authors of this meta-analysis, which included 30 studies, cited the presence of malignancy, a high Charlson score, and an increase in leukocyte values (1000/μL increase) and CRP (100 mg/L increase) as the most important independent risk factors for mortality [29]. Our research confirmed the previous conclusions that the increase in CRP represents an independent predictor of mortality in CDI patients (*p* = 0.015). The study conducted by Bednarska et al. showed that in the case of CRP concentration, the cut-off value associated with increasing the risk of death is 149 mg/L and that the risk of death increases by 50% with each 100 mg/L increase in CRP [41]. These conclusions can be interpreted by the fact that CDI patients with high CRP values have a more severe form of CDI, which is most likely caused by a strain of CD that produces a higher concentration of toxins, and therefore there is a higher risk of death. Accordingly, Hardt et al. emphasized that CRP is a better predictor for the development of severe forms of CDI than leukocytosis. These authors state that a CRP level of 250 mg/L at diagnosis predicts a probability higher than 50% for severe CDI [27]. On the other hand, high CRP values in patients with CDI may be a repercussion of the presence of more serious concomitant infections of other etiology, which puts these patients at additional risk of a poor outcome of the disease.

Our study highlighted certain similarities and differences in patients with CDI during the pandemic and prepandemic period. In order to recognize the risk group of patients in the early course of the disease and apply the appropriate treatment modality accordingly, our study highlighted some significant risk factors for the fatal outcome of the disease. Considering the excessive use of many medications during the pandemic, especially antibiotics, it is to be expected that a large number of patients will have a long-term imbalance of intestinal microbiota and reduced colonization resistance against CD as a consequence. Therefore, we can expect a further increase in CDI in the near future, so the proper management of antibiotics and prevention of the spread of CD remain indispensable measures in the control of infection. CDI prevention should primarily be aimed at hospitalized elderly people who receive antibiotics, as excessive use of antibiotics carries the risk of selection of highly resistant strains of CD. One of the problems also can be the fact that in the setting of CDI/SARS-CoV-2 coinfection it is difficult to monitor if diarrhea persist because of COVID-19 or reverse causation is present [44]. The World Health Organization has identified antimicrobial resistance as one of the biggest threats to global health [45]. Therefore, in the patient population that is the most at risk, the period of hospitalization should be shortened and the possibility of early de-escalation of antibiotic therapy should be considered in cases of need for concomitant antibiotic therapy.

Our study has several limitations. It is limited by its retrospective design and our inability to make a more detailed distinction between cases where CDI was the primary cause of death and cases where it was not because of the small number of microbiological analyses that confirmed concomitant infection in deceased patients. Furthermore, we did not have data on ribotyping in the study because PCR ribotyping of CD is not routinely performed in Serbia. Namely, clinical studies published in the last decade indicate an increase in more severe forms of CDI with higher mortality, which is partially attributed to the biological characteristics and pathogenicity of a specific strain of *Clostridioides difficile* known as ribotype 027 but also to some other ribotypes characterized by greater virulence and multiresistance to antibiotics [6]. Therefore, determining the CD ribotype in our patient population will be the goal of one of our future studies.

## 5. Conclusions

Patients with COVID-19 and CDI coinfection were significantly older, with more comorbidities, higher inflammatory markers, more extensive hypoalbuminemia, and a higher percentage of lethal outcomes than non-COVID-19/CDI patients. COVID-19 significantly changes the disease course of CDI and should be taken into consideration when managing patients with COVID-19 and CDI coinfection. Those patients should be treated with particular attention. 

## Figures and Tables

**Table 1 microorganisms-10-02284-t001:** Demographic characteristics, previous and concomitant medication use, laboratory parameters at the time of CDI diagnosis, as well as the outcome of treatment in the examined groups of patients.

Variable	CDI before COVID-19 (*n* = 304)	CDI and COVID-19 Coinfection (*n* = 387)	Test	*df*	*p* *
**Demographics**					
Age	65.6 ± 11.7	72.8 ± 9.4	−8.933	689	**<0.001**
Male gender	129 (42.6)	231 (59.8)	21.432	2	**<0.001**
**Comorbidities**					
Cardiovascular diseases	102 (33.6)	147 (38.0)	1.451	1	0.228
Diabetes mellitus	156 (51.3)	211 (54.5)	1.904	2	0.386
Gastrointestinal tract diseases	82 (27.0)	90 (23.3)	1.259	1	0.262
Chronic renal failure	71 (23.4)	131 (33.9)	9.065	1	**0.003**
Chronic liver disease	36 (11.8)	46 (11.9)	0	1	0.986
Malignancy	41 (13.5)	94 (24.3)	12.639	1	**<0.001**
Neurological diseases	62 (20.4)	71 (18.3)	0.46	1	0.498
Chronic obstructive lung disease	47 (15.5)	88 (22.7)	5.738	1	**0.017**
Autoimmune disorder	26 (8.6)	24 (6.2)	1.402	1	0.236
Hematological malignancy	10 (3.3)	19 (4.9)	1.111	1	0.292
Charlson comorbidity score	2.33 ± 1.34	3.06 ± 1.54	−6.584	689	**<0.001**
**Treatment before CDI**					
Antibiotics					
Cephalosporin of the III generation	137 (45.1)	181 (46.8)	0.199	1	0.655
Fluoroquinolone	119 (39.1)	175 (45.2)	4.897	2	0.086
Amoxicillin–clavulanic acid	17 (5.6)	28 (7.2)	0.736	1	0.391
Macrolide	26 (8.6)	149 (38.5)	80.748	1	**<0.001**
Other antibiotics	34 (11.2)	32 (8.3)	1.675	1	0.196
Antimotility drugs	16 (5.3)	7 (1.8)	6.314	1	**0.012**
Proton pump inhibitors	124 (40.8)	292 (75.5)	85.381	1	**<0.001**
Corticosteroid treatment	29 (9.5)	83 (21.4)	17.775	1	**<0.001**
Chemotherapy	25 (8.2)	57 (14.7)	6.889	1	**0.009**
**Laboratory findings at the moment of CDI diagnosis**					
Albumin (g/L)					
<25	37 (12.2)	76 (19.6)	10.968	2	**0.004**
25–30	109 (35.9)	102 (26.4)			
>30	158 (52.0)	209 (54.0)			
Creatinine (mmol/L)					
≤200	171 (56.2)	265 (68.5)	12.431	2	**0.002**
>200	133 (43.8)	122 (31.5)			
Leucocytes					
≤15 × 10^9^/L	156 (51.3)	217 (56.1)	1.551	1	0.213
>15 × 10^9^/L	148 (48.7)	170 (43.9)			
CRP (mg/L)	128.0 ± 80.14	133.4 ± 75.24	−0.923	689	0.357
**Concomitant treatment**					
Antibiotics	166 (54.6)	237 (100.0)	144.426	1	**<0.001**
Proton pump inhibitors	67 (22.0)	220 (100.0)	313.146	1	**<0.001**
Corticosteroids	25 (8.2)	55 (100.0)	226.515	1	**<0.001**
**Outcome**					
Lethal	20 (6.6)	114 (29.5)	57.01	1	**<0.001**

* Categorical variables are presented as *n* (%), while the differences between groups were assessed using the χ^2^-test. Continuous variables are presented as $\bar{x}$ ± SD, while the differences between groups were compared using the *t*-test according to the distribution; df: degree of freedom; CDI: *Clostridium difficile* infection; COVID-19: *Coronavirus disease 2019* (infection with the SARS-CoV-2 virus); CRP: C-reactive protein.

**Table 2 microorganisms-10-02284-t002:** Demographic characteristics, previous and concomitant medication use, and laboratory parameters at the time of CDI diagnosis in surviving and deceased patients in the prepandemic CDI group.

Variable	Prepandemic CDI Group—Recovered Patients (284)	Prepandemic CDI Group—Deceased Patients (20)	Test	*df*	*p* *
**Demographics**					
Age	65.2 ± 11.7	70.2 ± 11.5	1.824	302	0.069
Male gender	117 (41.3)	12 (60.0)	2.66	1	0.103
**Comorbidities**					
Cardiovascular diseases	92 (32.4)	10 (50.0)	2.598	1	0.107
Diabetes mellitus	144 (50.7)	12 (60.0)	15.304	2	**<0.001**
Gastrointestinal tract diseases	76 (26.8)	6 (30.0)	0.1	1	0.752
Chronic renal failure	65 (22.9)	6 (30.0)	0.528	1	0.467
Chronic liver disease	33 (11.6)	3 (15.0)	0.204	1	0.651
Malignancy	39 (13.7)	2 (10.0)	0.223	1	0.637
Neurological diseases	56 (19.7)	6 (30.0)	1.217	1	0.27
Chronic obstructive lung disease	41 (14.4)	6 (30.0)	3.463	1	0.063
Autoimmune disorder	25 (8.8)	1 (5.0)	0.345	1	0.557
Hematological malignancy	10 (3.5)	0 (0.0)	0.728	1	0.393
Charlson comorbidity score	2.29 ± 1.33	2.90 ± 1.45	1.976	302	**0.049**
**Treatment before CDI**					
Antibiotics					
Number of antibiotics before the infection	1.06 ± 0.48	1.85 ± 0.74	6.804	302	**<0.001**
Cephalosporin of the III generation	121 (42.6)	16 (80.0)	10.554	1	**0.001**
Fluoroquinolone	106 (37.3)	13 (65.0)	6.055	2	**0.048**
Amoxicillin–clavulanic acid	16 (5.6)	1 (5.0)	0.005	1	0.946
Macrolide	25 (8.8)	1 (5.0)	0.345	1	0.557
Other antibiotics	29 (10.2)	5 (25.0)	4.114	1	**0.043**
Antimotility drugs	15 (5.3)	1 (5.0)	0.003	1	0.957
Proton pump inhibitors	111 (39.1)	13 (65.0)	5.196	1	**0.023**
Corticosteroid treatment	24 (8.5)	5 (25.0)	5.93	1	**0.015**
Chemotherapy	23 (8.1)	2 (10.0)	0.09	1	0.765
**Laboratory findings at the moment of CDI diagnosis**					
Albumin (g/L)					
>25	253 (89.1)	14 (70.0)	6.366	1	**0.012**
≤25	31 (10.9)	6 (30.0)			
Creatinine (mmol/L)					
≤200	158 (55.6)	12 (60.0)	0.203	2	0.904
>200	125 (44.0)	8 (40.0)			
Leucocytes					
≤15 × 10^9^/L	156 (54.9)	0 (0.0)	22.566	1	**<0.001**
>15 × 10^9^/L	128 (45.1)	20 (100.0)			
CRP (mg/L)	116.67 ± 69.02	288.65 ± 48.77	10.944	302	**<0.001**
**Concomitant therapy**					
Antibiotics	146 (51.4)	20 (100.0)	17.797	1	**<0.001**
Proton pump inhibitors	63 (22.2)	4 (20.0)	0.052	1	0.82
Corticosteroids	24 (8.5)	1 (5.0)	0.295	1	0.587

* Categorical variables are presented as *n* (%), while the differences between groups were assessed using the χ^2^-test. Continuous variables are presented as $\bar{x}$ ± SD, while the differences between groups were compared using the *t*-test according to the distribution; df: degree of freedom; CDI: *Clostridium difficile* infection; COVID-19: *Coronavirus disease 2019* (infection with the SARS-CoV-2 virus); CRP: C-reactive Protein.

**Table 3 microorganisms-10-02284-t003:** Predictors of the lethal outcome in CDI before COVID-19.

Risk Factors	*p*	Odds Ratio	95% C.I. for Odds Ratio	
			Lower	Upper
Diabetes mellitus	0.820	0.833	0.173	4.010
Charlson comorbidity score	0.356	0.754	0.414	1.373
Number of antibiotics used before CDI onset	0.717	1.258	0.364	4.349
Cephalosporins of the third generations before CDI onset	0.993	1.009	0.136	7.459
Fluoroquinolones before CDI onset	0.485	0.552	0.104	2.935
CRP	**<0.001**	**1.033**	**1.017**	**1.049**
Corticosteroids before CDI onset	0.575	1.630	0.295	9.020
Concomitant antibiotic therapy	0.996	<0.001	<0.001	.
Without concomitant therapy	0.998	<0.001	<0.001	.
WBC more than 15 × 10^9^/L	0.528	2.380	0.161	35.173
Albumin less than 25 g/L	0.727	0.765	0.169	3.456
Concomitant PPI therapy	0.609	0.683	0.159	2.939

CDI: *Clostridium difficile* infection; COVID-19 (*Coronavirus disease 2019)*; C.I.: confidence interval; PPI: proton pump inhibitors; CRP: C-reactive protein; WBC—white blood cells.

**Table 4 microorganisms-10-02284-t004:** Demographic characteristics, previous and concomitant medication use, and laboratory parameters at the time of CDI diagnosis in surviving and deceased patients in the CDI- and COVID-19-coinfection group.

Variable	CDI and COVID-19 Coinfection—Surviving Patients (273)	CDI and COVID-19 Coinfection—Deceased Patients (114)	Test	*df*	*p **
**Demographics**					
Age	70.0 ± 6.8	80.2 ± 6.8	11.834	385	**<0.001**
Male gender	159 (58.2)	72 (63.7)	3.627	2	0.163
**Comorbidities**					
Cardiovascular diseases	92 (33.7)	55 (48.2)	7.223	1	**0.007**
Diabetes mellitus	135 (49.5)	76 (66.7)	9.613	1	**0.002**
Gastrointestinal tract diseases	53 (19.4)	37 (32.5)	7.664	1	**0.006**
Chronic renal failure	70 (25.6)	61 (53.5)	27.891	1	**<0.001**
Chronic liver disease	29 (10.6)	17 (14.9)	1.413	1	0.235
Malignancy	65 (23.8)	29 (25.4)	0.116	1	0.733
Neurological diseases	32 (11.7)	39 (34.2)	27.15	1	**<0.001**
Chronic obstructive lung disease	48 (17.6)	40 (35.1)	14.027	1	**<0.001**
Autoimmune disorder	19 (7.0)	5 (4.4)	0.916	1	0.339
Hematological malignancy	15 (5.5)	4 (3.5)	0.679	1	0.41
Charlson comorbidity score	2.70 ± 1.42	3.95 ± 1.45	7.827	385	**<0.001**
**Treatment before CDI**					
Antibiotics					
Number of antibiotics before the infection	1.26 ± 0.76	1.94 ± 0.93	7.399	385	**<0.001**
Cephalosporin of the III generation	100 (36.6)	81 (71.1)	38.275	1	**<0.001**
Fluoroquinolone	106 (38.8)	69 (60.5)	15.285	1	**<0.001**
Amoxicillin–clavulanic acid	25 (9.2)	3 (2.6)	5.103	1	**0.024**
Macrolide	103 (37.7)	46 (40.4)	0.233	1	0.629
Other antibiotics	11 (4.0)	21 (18.4)	21.96	1	**<0.001**
Antimotility drugs	5 (1.8)	2 (1.8)	0.003	1	0.959
Proton pump inhibitors	189 (69.2)	103 (90.4)	19.367	1	**<0.001**
Corticosteroid treatment	33 (12.1)	50 (43.9)	48.185	1	**<0.001**
Chemotherapy	44 (16.1)	13 (11.4)	1.423	1	0.233
**Laboratory findings at the moment of CDI diagnosis**					
Albumin (g/L)					
>25	240 (87.9)	71 (62.3)	33.477	1	**<0.001**
≤25	33 (12.1)	43 (37.7)			
Creatinine (mmol/L)					
<200	213 (78.0)	52 (45.6)	39.127	1	**<0.001**
>200	60 (22.0)	62 (54.4)			
Leucocytes					
≤15 × 10^9^/L	176 (64.5)	41 (36.0)	26.527	1	**<0.001**
>15 × 10^9^/L	97 (35.5)	73 (64.0)			
CRP (mg/L)	114.7 ± 60.7	178.4 ± 87.0	8.228	385	**<0.001**
**Type of concomitant therapy**					
Antibiotics	147 (100.0)	90 (100.0)	n.a		
Proton pump inhibitors	133 (100.0)	87 (100.0)	n.a		
Corticosteroids	28 (100.0)	27 (100.0)	n.a		

* Categorical variables are presented as *n* (%), while the differences between groups were assessed using the χ^2^-test. Continuous variables are presented as $\bar{x}$ ± SD, while the differences between groups were compared using the *t*-test according to the distribution; df: degree of freedom; CDI: *Clostridium difficile* infection; COVID-19: *Coronavirus disease 2019* (infection with the SARS-CoV-2 virus); CRP: C-reactive Protein.

**Table 5 microorganisms-10-02284-t005:** Predictors of the lethal outcome in the group of patients coinfected with CDI and COVID-19.

Risk Factors	*p*	Odds Ratio	95% C.I. for Odds Ratio	
			Lower	Upper
Age	**0.000**	**1.273**	**1.194**	**1.356**
Charlson comorbidity score	**0.019**	**1.352**	**1.051**	**1.739**
Proton pump inhibitors	0.620	1.284	0.478	3.449
Corticosteroids	0.252	1.806	0.657	4.962
Chemotherapy	0.080	0.413	0.153	1.112
WBC above 15 × 10^9^/L	0.732	0.843	0.317	2.242
Albumin ≤ 25 g/L	0.399	1.496	0.587	3.814
Creatinine > 200 mmol/L	0.131	1.702	0.853	3.396
CRP	**0.015**	**1.011**	**1.002**	**1.020**
Cephalosporin of the third generation	0.930	0.959	0.379	2.431
Fluoroquinolone	0.288	0.620	0.257	1.497
Amoxicillin–clavulanic acid	0.799	0.830	0.198	3.485

CDI: *Clostridium difficile* infection; COVID-19: *Coronavirus disease 2019* (infection with the SARS-CoV-2 virus); C.I.: confidence interval CRP: C-reactive protein; WBC—white blood cells.

## Data Availability

The data presented in this study are available on request from the corresponding author.

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
