# Peer review of "Clostridioides Difficile Infection before and during Coronavirus Disease 2019 Pandemic—Similarities and Differences"

_microorganisms, 2022, doi:10.3390/microorganisms10112284_

Round 1

Reviewer 1 Report

The aim of this paper was to compare the epidemiological, clinical characteristics and outcomes of Clostridioides difficile infection (CDI) during the COVID-19 pandemic compared to the pre-COVID era. It contributes to the understanding of the increase in morbidity and mortality of patients with CDI in the pandemic era as a probable consequence of the large number of hospitalized patients with comorbidities and excessive use of antibiotics.

The manuscript is clear, relevant for the field and presented in a well- structured manner. 

It is scientifically based. The tables are appropriate, and  they properly show the data which are easy to interpret and understand. Data are explained  appropriately throughout the manuscript. Statistical analysis is adequate.The conclusions  are consistent with the evidence and arguments presented.

It is necessary to more clearly define the two step algorithm in CDI diagnosis in the METHODS section.

There are some English language problems:

“Apstract” should be ABSTRACT

“deceased patients had bigger frequency of diabetes mellitus” should be HIGHER frequency…

Author Response

Respected,

Thank you very much for reading our manuscript, positive comments and useful advices for the improvement. 

Authors gave their best to improve the Methods section according to your advice. Also, we have corrected typing errors that you kindly noticed. 

Reviewer 2 Report

Clostridioides difficile infection (CDI) is a growing global threat for clinicians. While this disease has been affecting populations since ages, the advent of COVID-19 has definitely changed the dynamics and therapeutic strategies for CDI as well as other co-infection paradigms. In the current study the authors have done a very meticulous investigation of patient dynamics during pre- and post- COVID eras. Such studies are important to understand the scale of variation and severity, in disease manifestation and therapies. While reviewing this article I got a better understanding of the CDI/COVID-19 co-infection scenario and the combined effect of these diseases on patients.

Following are some minor comments:

1. Throughout the manuscript there are many instances where the references are cited in text as 'Author et al'. It would be more appropriate to mention these as 'Author and co-workers'.

2. Many numerical values with decimals have been shown with a comma. It would be appropriate to change this to a standard format.

Author Response

Thank you very much for evaluation of our manuscript and positive comments. Authors gave their best to make corrections of the way of presenting results, according to your advice. 

Reviewer 3 Report

The authors present an interesting view of clinically relevant outcomes- CDI and COVID. The data presented is relevant and findings interesting. Manuscript methods are appropriate and results support the conclusion/ discussion. I would suggest that the authors simplify the tables. There is currently too much data presented in each table and, while data is relevant, a more streamlined presentation of this data is needed. 

Author Response

Thank you for careful evaluation of our manuscript and positive comments. About tables, we have tried to make it more readable. 

Reviewer 4 Report

I read this manuscript with interest, however, there are several shortcomings and issues with the present report.

Specific comments:

1. As per the journal's guidelines to authors, the abstract should be a total of about 200 words maximum. The abstract should be a single paragraph and should follow the style of structured abstracts, but without headings.

2. "... around 60% of clinical CD strains are already reported as MDR in European hospitals" - at least a citation is necessary here.

3. In the introduction, it should also be mentioned that increasingly, C. difficile infections demonstrate an emerging pattern of resistance to available treatment and recurrence after an initial episode (citation: pubmed.ncbi.nlm.nih.gov/30391527). 

4. "Ethics Committee of the Hospital" - please provide the actual IRB study/approval number.

5. Scientific names such as "Clostridioides difficile" should be italicized as per convention.

6. It is unclear how the cases were identified by the study investigators. Was it through the use of electronic databases/ICD codes? Please specify.

7. What was the local prevalence of CDI and SARS-CoV-2 in the area? These baseline rates should be more clearly stated.

8. Probably, the most interesting finding of the study by Lewandowski et al. lies in the observation that the COVID-19-related involvement of the gastrointestinal tract is an independent risk factor for CDI. The reason for this association could depend on the alteration of the gut microbiome caused by SARS-CoV-2. A growing body of evidence has shown that COVID-19 is associated with a disruption of the human gut microbiome, specifically, an increase in the number of opportunistic pathogens along with a decrease in beneficial commensals, which can persist in most patients even several weeks following the clearance of SARS-CoV-2.

9. When CDI is present as a co-infection with COVID-19, CDI therapy can be difficult to monitor if diarrhoea persists because of COVID-19. Of course, reverse causation can also be an issue here as patients can contract both CDI and COVID-19 in hospital settings especially if hospitals are hotspots for COVID infections (citation: pubmed.ncbi.nlm.nih.gov/35043103).

10. "... a decrease in the acidity of gastric juice" - odd expression, suggest to reword this in a more scientific manner, please.

11. Although some patient variables were studied, it would be difficult to arrive at firm conclusions because COVID patients would likely have more antibiotic exposure (having presented with fever +/- respiratory symptoms) compared to non COVID patients and HD/ICU admission is a major risk factor- and this needs to be controlled for.

Author Response

Thank you for the careful reading of our manuscript and useful suggestions. 

Authors gave their best to improve the manuscript according to your recommendations, point by point.

  1. abstract is restructured and shortened
  2. citation added
  3. we have added that fact in the Introduction section, with the belonging reference
  4. IRB number is added into the Methods section
  5. All Latin terms are now in italic
  6. We tried to be more specific in Methods section by explaining that data were taken from electronic healthcare records. 
  7. Authors tried to get the information about prevalence of CDi and SARS-CoV-2 infection from Institute for public heath - department for Epidemiology. For 2022, those data are still in processing and not available in this moment. 
  8. We have added that fact in the Introduction section.
  9. Citation added
  10. corrected
  11. "Although some patient variables were studied, it would be difficult to arrive at firm conclusions because COVID patients would likely have more antibiotic exposure (having presented with fever +/- respiratory symptoms) compared to non COVID patients and HD/ICU admission is a major risk factor- and this needs to be controlled for."  - exactly those facts were one of the reasons for designing the study with the concept of evaluating risk factors separately in precovid and covid era. 

Once again, thank you for careful reading and useful suggestions. 

  1.  

Round 2

Reviewer 4 Report

Thank you for the revisions.